# Effects of Dietary Microcapsule Sustained-Release Sodium Butyrate on the Growth Performance, Immunity, and Gut Microbiota of Yellow Broilers

**DOI:** 10.3390/ani13233598

**Published:** 2023-11-21

**Authors:** Zhenglie Dai, Xiuxi Wang, Yulan Liu, Jinsong Liu, Shiping Xiao, Caimei Yang, Yifan Zhong

**Affiliations:** 1College of Animal Science and Technology, College of Veterinary Medicine, Zhejiang Agricultural and Forestry University, Hangzhou 311300, China; mmddaa0323@163.com (Z.D.); 15261825191@163.com (X.W.); 2Zhejiang Vegamax Biotechnology Co., Ltd., Huzhou 313300, China; yulanflower@126.com (Y.L.); 13906510961@163.com (J.L.); xiao1974920@126.com (S.X.); yangcaimei2012@163.com (C.Y.)

**Keywords:** microcapsule sustained-release sodium butyrate, yellow broiler, gut health, immunity, gut microbiota

## Abstract

**Simple Summary:**

The supplementation of sodium butyrate (SB) has proven to be an effective way to provide extra butyrate to improve the homeostasis and gut health of broilers and consequently enhance their performance. However, the rapid release of SB in the gastrointestinal system leads to lower efficiency in poultry production. Sustained-release sodium butyrate (MSSB) has shown potential, with a lower dosage required and a better effect on performance. A comparison of rapid SB absorption and MSSB in terms of performance, serum parameters, and gut microbiota is poorly demonstrated in the literature, especially in yellow broilers, a primary source of the live poultry trade in China. Our study found that, overall, MSSB contributes to improved performance, immunity, and gut health. Specifically, the supplementation of MSSB lowered the feed:gain ratio and increased the IgA, isovalerate, and villus height of yellow broilers. Alterations in gut microbiota, including the predicted composition and function, have also been observed, namely, in *Clostridia* UCG-014, *Bacilli* RF39, and *Oscillospiraceae* UCG-005. The mediation analysis revealed a causal effect between *Clostridia* UCG-014 in the gut and serum IgA, with the tryptophan metabolism appearing to be a key mediator in this relationship. This study shows that MSSB can improve the performance and enhance the immunity of yellow broilers. Our results are significant because they reveal the beneficial effects together with the potential mechanism of MSSB in yellow broilers.

**Abstract:**

The beneficial effects of butyric acid in poultry production are well documented, while the relationship between sodium butyrate (SB) and microcapsule sustained-release sodium butyrate (MSSB), especially in yellow broilers, remains poorly investigated. This study was designed to elucidate the function as well as the potential mechanisms of SB and MSSB in enhancing health in yellow broilers. In total, 360 one-day-old yellow broilers were allocated to three treatment groups. The control group (CON) received a basic diet, while the SB group was provided with 1000 mg/kg of sodium butyrate (SB), and the MSSB received microcapsule sustained-release sodium butyrate (MSSB), all over a period of 56 days. Compared to the CON group, the dietary supplementation of both SB and MSSB showed a lower feed:gain ratio (*p* < 0.01). No significant (*p* > 0.05) difference in antioxidant capacity was observed between the three groups. We observed significantly higher levels (*p* < 0.05) of immunoglobulins and a reduction in concentrations in both the SB and MSSB groups compared to the CON group. Furthermore, both SB and MSSB induced alterations in the diversity, structure, and function of gut microbiota. MSSB demonstrated even more pronounced beneficial effects than SB, particularly in regard to the serum IgA level (*p* = 0.05), cecal isovalerate concentration (*p* < 0.05), and villus height (*p* < 0.01). The sequencing of the gut microbiota revealed that MSSB led to a significant increase in the relative abundance of *Clostridia* UCG-014, *Bacilli* RF39, and *Oscillospiraceae* UCG-005. Predictions of bacterial function indicated changes in KEGG pathways, including an enrichment of tryptophan metabolism (ko00380), and a reduction in fructose and mannose metabolism (ko00051), chloroalkane and chloroalkene degradation (ko00625), and naphthalene degradation (ko00626) in yellow broilers fed with MSSB. Among these, the mediation analysis revealed a causal effect between the *Clostridia* UCG-014 in the gut and serum IgA, with tryptophan metabolism being a key mediator in this relationship. Our results suggest that dietary MSSB can improve the growth performance, immunity, and gut microbiota of yellow broilers. MSSB increased the abundance of *Clostridia* UCG-014 and activated the tryptophan metabolism pathway (ko00380), contributing to IgA levels in yellow broilers through this mechanism.

## 1. Background

Poultry production plays a crucial role in supplying high-quality and cost-effective animal protein on a global scale. However, in the modern poultry industry, broilers with a rapid growth rate are at risk of increased stress and immune challenges [1]. The composition and function of gut microbiota have already been linked to the gut barrier function, growth performance, and immunity of the host [2,3]. Thus, it is necessary to maintain the homeostasis of gut microbiota in broilers.

As one of the growth promoters and immune function regulators in the field of animal nutrition, sodium butyrate (SB) has attracted global interest [4,5]. Butyrate acts as a key short-chain fatty acid in the gut and serves as primary fuel for the metabolism of colonocytes, while also improving the integrity of epithelial tissue, alleviating the inflammation of mucosa, and stimulating the absorption of electrolytes [6,7]. Furthermore, dissociated butyric acid can penetrate freely into the cytoplasm, inhibiting the replication of DNA and dissociating the nutrient transport system from bacteria, leading to a broad-spectrum antibacterial effect of SB [8]. However, a study of pigs showed that the supplementation of uncoated SB can be dissociated in the anterior section before it reaches the lower gut, such as the small intestine [9], limiting its impact on gut microbiota and barrier function. Additionally, the strong odor of SB might exert an adverse effect on the feed intake of broilers [10]. Thus, the microcapsule sustained-release of SB (MSSB) was able to effectively deliver butyrate throughout the entire gastrointestinal tract, contributing to its beneficial role in the improvement of gut microbiota and maintenance of gut health [11]. As a result, the pretreatment of a microcapsule could lower the usage of SB in poultry production and exert greater beneficial effects than uncoated SB.

In China, yellow broilers are the primary source of the live poultry trade, with a rapid growth rate, accounting for 40% of the total chicken meat production in China [12]. Numerous studies have shown the beneficial effects of SB on broiler production parameters, including feed intake, weight gain, and the feed conversion ratio [13]. The supplementation of SB for regulating gut microbiota, promoting epithelium development, and enhancing the immune function of broilers are well documented [14]. Nevertheless, there remains a need for comprehensive data on the effect of SB and MSSB on the immune function of yellow broilers. The glutathione peroxidase (GSH-Px), total antioxidant capacity (T-AOC), and superoxide dismutase (SOD) are important parameters for assessing oxidative stress [15]. Immunoglobulin A (IgA), immunoglobulin M (IgM), and immunoglobulin Y (IgY) in serum are the main indicators of immune response in broilers [16]. Also, interleukin, including interleukin-6 (IL-6) and interleukin-1*β* (IL-1*β*), together with tumor necrosis factor-α (TNF-α) are the key markers of inflammation and have diverse physiological functions in host health [17]. Further investigation is required to reveal the potential causal relationship between gut microbiota and immunity in yellow broilers. As such, this study was designed to compare the effect of SB and MSSB on the growth performance, immune function, together with the bacterial composition and function in the gut, of newborn yellow broilers, aiming to demonstrate the potential mechanisms of MSSB applied as a feed additive in these broilers.

## 2. Materials and Methods

### 2.1. Experimental Design and Animal Management

The MSSB was obtained from Zhejiang Vegamax Biotechnology Co., Ltd., Huzhou, China. The animal study was approved by the Animal Care and Use Committee of Zhejiang Agricultural and Forestry University (ethical code: ZAFU20230262). The MSSB was coated with polymer enteric material, containing 40% SB, and obtained from Zhejiang Vegamax Biotechnology Co., Ltd.

In total, 360 one-day-old yellow broilers were randomly selected and divided into 3 treatments with 8 replicates and 15 birds/replicate. The yellow broilers were fed a basal diet (CON), CON diet supplemented with 1000 mg/kg sodium butyrate (SB) of diet, or CON diet supplemented with 1000 mg/kg microcapsule sustained-release sodium butyrate (MSSB) of diet. The dosage of MSSB was based on a previous study [18].

Yellow broilers were raised in a house with a length of 2 m, a width of 1.5 m, and a height of 2 m. Each replication was raised in a cage under an age-appropriate temperature, light regime, and humidity, with free access to food and water, and vaccinated according to the protocol. The basic diet was formulated with reference to the recommendation of the “Nutritional Requirements of Chinese Yellow Feather Broilers” (NY/T 3645-2020) [19] and “Feeding Standard of chicken” (NY/T 33-2004) [20], and its ingredients and nutrient levels are shown in Table 1.

### 2.2. Sample Collection

At 56 d of age, 1 bird per replicate close to the average body weight was selected and weighed. A total of 10 mL fresh blood was collected from the jugular vein and centrifuged at 5000 r/min for 10 min at 4 °C. The isolated serum was collected in a tube and stored at −80 °C for further analysis. The broilers were slaughtered and about 1 cm of the middle portion of the jejunum was fixed in 4% paraformaldehyde and stored at 4 °C for histomorphology analysis. The digesta of cecum were collected in sterile freezer tubes and stored at −80 °C for further analysis.

### 2.3. Growth Performance

The BW of 1- and 56-day-old broilers was measured and the feed consumption of yellow broilers was recorded for each replicate during the experimental period. The average daily weight gain (ADG), average daily feed intake (ADFI), and feed:gain ratio (F:G) throughout the experimental period were calculated for each bird.

### 2.4. Analysis of Serum Parameters

The glutathione peroxidase (GSH-Px), total antioxidant capacity (T-AOC), and superoxide dismutase (SOD) in the serum were detected with reagent kits according to the instructions of the manufacturer (Nanjing Jiancheng Bioengineering Institute, Nanjing, China). The serum cytokines including immunoglobulin A (IgA), immunoglobulin M (IgM), immunoglobulin Y (IgY), interleukin-6 (IL-6), interleukin-1*β* (IL-1*β*), and tumor necrosis factor-α (TNF-α) were detected using an ELISA kit according to the instructions of the manufacturer (Nanjing Angle Gene Co., Ltd., Nanjing, China). The sensitivity of the kits to GSH-Px was from 20 U to 330 U. The sensitivity of the kits to T-AOC was from 0.2 U/mL to 55.2 U/mL. The sensitivity of the kits to SOD was from 5.0 U/mL to 122.1 U/mL. The sensitivity of the kits to IgA was from 1.25 μg/mL to 75 μg/mL. The sensitivity of the kits to IgM was from 0.1405 μg/mL to 11.25 μg/mL. The sensitivity of the kits to IgG was from 7.5 μg/mL to 450 μg/mL. The sensitivity of the kits to IL-1β was from 5 pg/mL to 300 pg/mL. The sensitivity of the kits to IL-6 was from 5 pg/mL to 300 pg/mL. The sensitivity of the kits to TNF-α was from 5 pg/mL to 300 pg/mL.

### 2.5. Gut Morphology Analysis

Samples of 0.5 cm jejunum were fixed with 4% formaldehyde (Aladdin, Shanghai, China) solution for at least 24 h, dehydrated, embedded in paraffin (4 µm), and stained with hematoxylin–eosin. Images of sections were mounted and photographed with a fluorescence microscope (DS-FI2 camera, EclipseCi, Nikon, Japan). The villus height (VH) and crypt depth (CD) were measured at 40× magnification at 10 fields per sample. The VH was measured from the top of the villi connected to the crypt, and the VCD was defined as the depth of the crypt between two adjacent villi [21]. From these measurements, the ratio of VH to CD was calculated.

### 2.6. Short-Chain Fatty Acid (SCFA) Analysis

The SCFAs including the acetate, propionate, isobutyrate, butyrate, isovalerate, and valerate in cecum digesta were quantified by gas chromatography [22]. In brief, 0.5 g of cecal contents was taken into the centrifuge tube, combined with ddH_2_O, and centrifuged. The supernatant was taken and mixed with 25% metaphosphoric acid and centrifuged at 4 °C, 12,000 r/min for 10 min. The supernatant was filtered and transferred for testing. An Agilent 6890N gas chromatography system (Agilent, Santa Clara, CA, USA) was applied for high-performance liquid chromatography (HPLC).

### 2.7. 16S rRNA Gene Sequencing

The cecal microbiota of yellow broilers were sequenced. Briefly, primers 341F (5′-CCTACGGGRSGCAGCAG-3′) and 806R (5′-GGACTACVVGGGTATCTAATC-3′) targeting V3-V4 region [23] were applied for the amplification of the bacterial 16S rRNA gene. PCRs were performed, and the Qubit was used for the construction and assessment of the library. Finally, the library was sequenced on an Illumina MiSeq platform (Illumina, San Diego, CA, USA) using PE 2 × 250 in Majorbio Bioinformatics Technology Co., Ltd. (Shanghai, China).

QIIME 2 (version 2020.8, https://qiime2.org, accessed on 1 January 2018) was used for demultiplexing (Q2-DEMUX) and processing the raw fastq files. The DADA2 pipeline was applied for filtering, dereplication, chimera identification, and merging paired-end reads, which produced 6684 amplicon sequence variants (ASV). The SILVA 138 database was used for the taxonomic classification of representative sequence sets. The alpha diversity, including Shannon, Simpson, Ace, and Chao 1 indices, was calculated using QIIME2. The beta diversity was evaluated using the Bray–Curtis distances in QIIME2. PICRUST2 was applied for the prediction of bacterial function, including the KEGG pathway (KEGG, v77.1), modules, and CAZymes [24].

### 2.8. Statistical Analysis

The data from the Shapiro–Wilk test were calculated via one-way ANOVA followed by Fisher’s LSD test, including the performance, concentration of serum parameters, SCFAs, and gut histomorphology in GraphPad Prism 8.0. The Breusch–Pagan test was applied to assess the homoscedasticity. Linear discriminant analysis (LDA) effect size (LEfSe) [25] was performed in R with the “lefser” package to screen the differential bacteria in the three groups. Principal coordinate analysis (PCoA) was visualized and PERMANOVA was applied to identify the structures of gut microbiota between the three groups in R software (v3.3.1). Spearman correlation analysis between differential genera, KEGG pathways, and serum parameters was performed in R and visualized in Cytoscape (v3.2.1). For genera associated with both KEGG pathways and IgA, bi-directional mediation analysis with interactions between the mediator and outcome was performed with the “mediation” package in R software. The results were displayed in the form of the boxplot and standard error of mean values, where *p* < 0.05 indicates statistical significance.

## 3. Results

### 3.1. Feed Intake and Performance

As shown in Table 2, the average daily gain (ADG, g) of the yellow broilers showed no difference (*p* = 0.13) between groups, which was 34.44, 37.42, and 38.27 in CON, SB, and MSSB, respectively. The average daily feed intake (ADFI, g) was 69.56, 73.54, and 74.50, respectively, and showed a significant difference (*p* = 0.02) between the three groups. The feed:gain ratio (F:G) of the yellow broilers for the CON, SB, and MSSB groups was 2.37, 2.23, and 2.25, respectively. The F:G between the three groups showed significant difference (*p* < 0.01). Both the SB and the MSSB groups showed significantly (*p* < 0.01) lower F:G than the CON group, while no significant difference (*p* = 0.83) was observed between SB and MSSB.

### 3.2. Serum Parameters

The serum parameters between the three groups were also detected. As shown in Figure 1A, the antioxidant index showed no difference between the three groups, that is, GSH-PX (*p* = 0.11), T-AOC (*p* = 0.06), and SOD (*p* = 0.88). In terms of immunity (Figure 1B), the administration of both SB and MSSB showed higher levels of IgA, IgM, and IgY (*p* < 0.01) than those in the CON group. Based on a comparison between SB and MSSB, the IgA showed a higher trend (*p* = 0.05) in MSSB than that in SB, while the IgM (*p* = 0.34) and IgY (*p* = 0.13) showed no difference between any two groups. Compared to the CON group, a lower (*p* < 0.01) level of inflammatory cytokines including IL-6, IL-1*β*, and TNF-α in the SB and MSSB groups was observed (Figure 1C). Moreover, the IL-6 (*p* = 0.51), IL-1*β* (*p* = 0.48), and TNF-α (*p* = 0.38) showed no difference between the SB and MSSB groups.

### 3.3. Gut SCFAs and Morphology

The gut SCFAs were also detected (Figure 2A), and the isovalerate in MSSB showed a significantly (*p* < 0.05) higher level than in the CON and SB groups. Apart from the isovalerate, other SCFAs including the acetate (*p* = 0.26), propionate (*p* = 0.12), isobutyrate (*p* = 0.19), butyrate (*p* = 0.09), and valerate (*p* = 0.23) showed no difference between the three groups. The gut morphology was also investigated (Figure 2B,C), the villus height (VH) of yellow broilers in the MSSB group was significantly higher (*p* < 0.01) than that in the CON and SB groups, while the villus height of the CON and SB groups showed no difference (*p* = 0.92). The crypt depth (CD, *p* = 0.48) and ratio of VH to CD (*p* = 0.10) showed no difference between the three groups. As shown in Figure 2C, the denser villus in the MSSB group was denser than that in the CON and SB groups.

### 3.4. Profiles of Gut Microbiota

The gut microbiota of yellow broilers was sequenced and a total of 1472 ASVs with a relative abundance over 0.1% in at least one group was achieved (Appendix A). The composition and structure of the gut microbiota between CON, SB, and MSSB was also illustrated. At the phylum level, nine phyla were assigned, namely, Firmicutes, Bacteroidota, Actinobacteriota, Verrucomicrobiota, Cyanobacteria, Campilobacterota, Desulfobacterota, Proteobacteria, and Synergistota. Among these, Firmicutes and Bacteroidota were the predominate phyla in three groups (Appendix A). At the genus level, there are 35 genera with a relative abundance over 1% in at least one group (Appendix A). The *Bacteroides*, *Christensenellaceae* R7 group, and unclassified_f_*Lachnospiraceae* were the predominant genera in all groups.

The alpha diversity index of gut microbiota was calculated (Figure 3A), and both SB and MSSB showed significantly higher (*p* < 0.01) richness and diversity but lower (*p* < 0.01) evenness than that in the CON group, while no significant difference was observed between the SB and MSSB groups. The PCoA plot was illustrated to reveal the structures of gut microbiota (Figure 3B), with a clear separation (*p* < 0.05) between the CON and treatment groups (SB and MSSB), while a similar structure between the SB and MSSB groups (*p* = 0.09) was observed. When LefSe was applied to reveal the differential genera between the three groups (Figure 3C,D), six genera with an LDA value over 4.0 were revealed, namely, *Lactobacillus*, *Streptococcus*, *Faecalibacterium*, *Clostridia* UCG-014, *Oscillospiraceae* UCG-005, and *Bacilli* RF39.

### 3.5. Difference between Functions of Gut Microbiota

The functions of gut microbiota were predicted and are shown in Figure 4. The KEGG pathway (Figure 4A), module (Figure 4B), and CAZyme (Figure 4C) profiles of the gut microbiota in the SB and MSSB groups showed a significant (*p* < 0.05) difference compared with the CON group. The correlations between the functions of gut microbiota and differential genera (Figure 4D) or serum parameters (Figure 4E) were established. Mediation analysis was carried out to investigate the links between gut microbiota, pathways, and serum parameters (Figure 4F). For the genera that were associated with both pathways and IgA (*p* < 0.05), bi-directional mediation analysis was applied to evaluate the effects of gut microbiota and serum parameters specifically altered in MSSB. Most linkages were related to the impact of *Clostridia* UCG-014 on serum IgA levels via ko00051 (fructose and mannose metabolism), ko00380 (tryptophan metabolism), ko00625 (chloroalkane and chloroalkene degradation), or ko00626 (naphthalene degradation).

## 4. Discussion

This study aimed to compare the effects of sodium butyrate in different forms on the performance, immunity, and gut microbiota of yellow broilers. Sodium butyrate has been widely applied as a feed additive to improve the performance and health not only of broilers [26], but also of turkeys [27], pigs [28,29], and ruminants [30,31]. MSSB contains only 40% sodium butyrate but exerts better effects in yellow broilers. In our study, MSSB lowered the F:G ratio of yellow broilers, which is consistent with previous studies [13,32,33]. No significant differences in performance, including ADF and ADFI, were observed between the CON, SB, and MSSB groups.

### 4.1. Both SB and MSSB Could Exert Immunomodulatory Effects

During the early stages of growth, broilers are particularly susceptible to stimulation by external factors, especially when the immune function is not fully established. This vulnerability can lead to compromised health outcomes [34]. In our study, the serum parameters were detected, and the antioxidant capacity such as GSH-PX, T-AOC, and SOD also showed no difference in broilers supplied with SB or MSSB. Nonetheless, both SB and MSSB could exert beneficial effects via strengthening the immune function of broilers. Elevated levels of serum immunoglobulin levels including IgA, IgY, and IgM were observed in the yellow broilers administered with SB and MSSB. IgA is the most abundant antibody isotype in the body and the response of T-cell-independent IgA could be promoted by butyrate [35]. A recent study showed that maternal supplementation with dietary SB substantially increased the serum IgA levels both in breeders and offspring in Ross308 female broilers [4]. A study of pigs also showed that SB could contribute to a higher IgA level in colostrum [36]. A recent study showed that circulating levels of IgM and IgY were higher in the fast-growth broilers than those in slow-growth broilers, which might be biomarkers to discriminate performance [37]. Elevated levels of IgM and IgY in broilers of SB and MSSB were also observed in our study, suggesting the growth potential of broilers fed with SB or MSSB. The immunomodulatory effects of sodium butyrate were also observed, not only in immunoglobin, but also in cytokines, including IL-6, IL-1*β*, and TNF-α, by modulating the gut microbiome [6,38]. SCFAs are the main metabolites produced in the gut [39] and are considered to alleviate inflammation [40]. A loss of butyrate is often associated with increased proinflammatory cytokines [41]. The reduction levels of IL-6, IL-1*β*, and TNF-α were observed in the broilers fed with SB or MSSB, which further proved the positive influence of dietary supplement sodium butyrate on the broilers. Consistent with previous studies, the dietary supplementation of protected SB decreased the proinflammatory cytokines IL-6 and TNF-α in Arbor Acres broilers at an early stage [21]. To better illustrate the exact mechanisms of the immunomodulatory effects of butyrate on yellow broilers, further study is warranted.

### 4.2. MSSB Contributes to Further Beneficial Effects Compared to SB

During the development of an infant’s gut, butyrate plays an important role via the interaction with the immune system and is linked to health in later life [42]. MSSB can ensure that it works in the intestinal tract without being absorbed in the stomach, which could better contribute to gut development than SB. In our study, the butyrate level showed no significant difference between the three groups, possibly because of the sustained-release treatment and the rapid absorption of butyrate in the gut [43]. Similar butyrate concentration in the gut of chickens was observed in chickens treated with coated SB and in the control group [21]. Nonetheless, a greater villus height was observed in the MSSB than that in the CON and SB groups, while no significant difference was found between the CON and SB groups. The uncoated SB is dissociated in the upper gastrointestinal tract; as a result, a considerable quantity of SB cannot be utilized by enterocytes due to low concentrations, leading to a lower villus length in SB than in MSSB. The increased villus length in MSSB might also contribute to the decrease in F:G in yellow broilers.

A previous study reported a dose–response relationship between the sodium butyrate levels and concentrations of DNA, RNA, and protein in the gut mucosa of chickens [44]. Greater villus height is usually associated with a promotion of digestibility and absorption by the gut, which, in turn, enhances the health status of the livestock [45]. On the other hand, among the differential immunoglobulins and cytokines, a difference in the IgA level was observed not only between the control and treatment groups, but also between the SB and MSSB groups, suggesting that IgA might be the main target of MSSB. Previous studies suggested that the concentrations of SCFAs in Luminal, especially butyrate, could induce the differentiation of T cells both in vitro and in vivo [46]. The IgA is the most abundant antibody isotype produced and the secretion of IgA in the gut could provide critical protection against pathogens and bacteria, modulating the gut microbiota composition and enforcing homeostasis in vertebrates [47].

### 4.3. Supplementation of SB and MSSB Shaped the Gut Microbiota Profiles in Early Life

It has been extensively suggested that commensal bacteria usually play a critical role in the interaction between the supplementation of butyrate and host immunity [48,49]. The gut microbiota of broilers has been illustrated, and a higher diversity and differential structure of microbiota in the SB and MSSB groups were observed compared to those in CON. Human studies revealed that the lower microbial diversity of infants might lead to a disordered microbial community and a delayed immune system [50]. Moreover, changes in the gut microbiome and immunity, such as metabolic syndromes, can be characterized by the low bacterial diversity and lowered levels of butyrate-producing bacteria [51]. A study aiming to describe the temporal changes of the gut microbiota in broilers suggested that gut homeostasis might be affected by a loss of bacteria [52], indicating the importance of preserving microbial diversity. Thus, the dietary supplementation of SB or MSSB could contribute to the mutation and diversity of gut microbiota in broilers. For example, LEfSe showed that broilers fed with SB or MSSB decreased the relative abundance of *Streptococcus*. The genus *Streptococcus* comprises commensals with pathogenic potential or true pathogens [53], which have been associated with the local and systemic inflammation of the host [54,55] and lead to the growth of depression in broilers as well as increased mortality without obvious clinical signs [56]. On the other hand, dietary MSSB could exert beneficial effects via modulation not only on composition, but also on the function of gut microbiota, especially *Clostridia* UCG-014, *Bacilli* RF39, and *Oscillospiraceae* UCG-005. *Clostridia* UCG-014 has been reported to be positively associated with bacterial diversity and barrier function in healthy individuals [57,58]. *Bacilli* RF39 is one of the prevalent members and is able to produce metabolites, including acetate and hydrogen, but lacks the ability for other highly conserved metabolic pathways [59]. The supplementation of MSSB enables *Bacilli* RF39 to utilize more substrate for growth, and that might be why elevated levels of *Bacilli* RF39 occur in the lower gut of broilers. *Oscillospiraceae* are one of the major gut lineages that have been regularly correlated with health biomarkers, such as bacterial richness, and are considered to show anti-inflammatory effects [60,61]. Although difficult to culture temporarily, *Oscillospiraceae* UCG-005 has been reported to be positively correlated with SCFA concentration and negatively correlated with inflammation in several studies [62,63,64].

### 4.4. MSSB Potentially Improves the IgA Level via Tryptophan Metabolism in Clostridia UCG-014

The function of microbiota was predicted and correlated with the differential genera and serum parameters. Among the correlations between the differential genera, pathways, and serum parameters, mediation analysis was applied to reveal the putative causal relationships [65] triggered by the dietary MSSB. As one of the potential targets of MSSB, the IgA and differential genera, including *Clostridia* UCG-014, *Bacilli* RF39, and *Oscillospiraceae* UCG-005, together with the KEGG pathways were applied for bi-directional analysis. The differential KEGG pathway includes tryptophan metabolism (ko00380), fructose and mannose metabolism (ko00051), chloroalkane and chloroalkene degradation (ko00625), and naphthalene degradation (ko00626) as the potential mediators of the interaction between *Clostridia* UCG-014 and the IgA level in broilers fed with MSSB. As a dominant class of commensal bacteria, *Clostridia* can induce colonic regulatory T cells, and has a central role in the suppression of inflammation [66]. *Clostridia* UCG-014 was evidently more reduced in individuals with gut inflammation than in healthy individuals, and is the major source of tryptophan metabolites. Previous studies have suggested that tryptophan metabolites could regulate intestinal homeostasis via the tryptophan-AhR pathway [67,68]. Moreover, 16S rRNA sequencing combined with targeted metabolomics demonstrated that an increased abundance of *Clostridia* UCG-014 could improve the gut barrier function through the activation of tryptophan metabolism in mice cecal [69]. The disturbance of SCFAs as well as the tryptophan in the gut leading to the disorder of the IgA [70] suggests a relationship between tryptophan metabolism and the IgA level. Apart from tryptophan metabolism, the lower level of pathways including chloroalkane and chloroalkene degradation and naphthalene degradation were observed in MSSB. The degradation of exogenous substances such as chloroalkane and chloroalkene and naphthalene in gut microbiota could lead to the impairment of glomerular filtration and increase the burden on kidneys [71,72,73]. Enrichment studies and metagenomic analysis have suggested the potential capacity of uncultivable *Clostridium* members in the degradation of chloroalkane, chloroalkene, and naphthalene [74,75]. Despite a further demonstration of the underlying mechanisms, the supplementation of MSSB lowered the pathway, including chloroalkane and chloroalkene degradation and naphthalene degradation in *Clostridia* UCG-014, further promoting the immunity of yellow broilers.

## 5. Conclusions

In this study, both SB and MSSB improved feed efficiency, including the ADFI and F:G, and promoted serum immunoglobin levels, including IgA, IgM, and IgY, while alleviating cytokines such as IL-6, IL-1*β*, and TNF-α. Bacterial diversity was elevated, and genera such as *Clostridia* UCG-014 and the tryptophan metabolism of the gut microbiota were altered. The alteration of gut microbiota and function potentially contributed to immunity in broilers. MSSB reduced the usage of sodium butyrate and exerted better effects than SB, especially the serum IgA level, isovalerate concentration, and villus height in the gut. The results above suggested that MSSB could improve the performance and immunity of IgA via modulating the gut microbiota *Clostridia* UCG-014 and tryptophan metabolism in yellow broilers.

## Figures and Tables

**Figure 1 animals-13-03598-f001:**
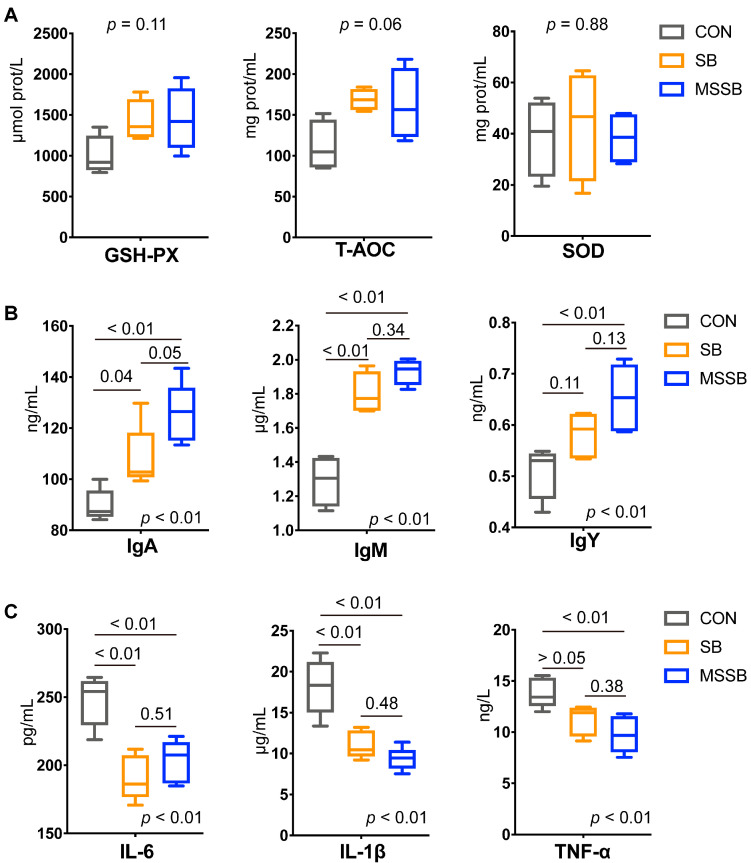
The serum parameters of yellow broilers between CON, SB, and MSSB groups. (**A**) The antioxidant index including GSH-PX, T-AOC, and SOD. (**B**) The immunity including IgA, IgM, and IgY. (**C**) The inflammatory cytokines including IL-6, IL-1*β*, and TNF-α between the three groups. Abbreviations: CON, control; SB: sodium butyrate; MSSB: microcapsule sustained-release sodium butyrate.

**Figure 2 animals-13-03598-f002:**
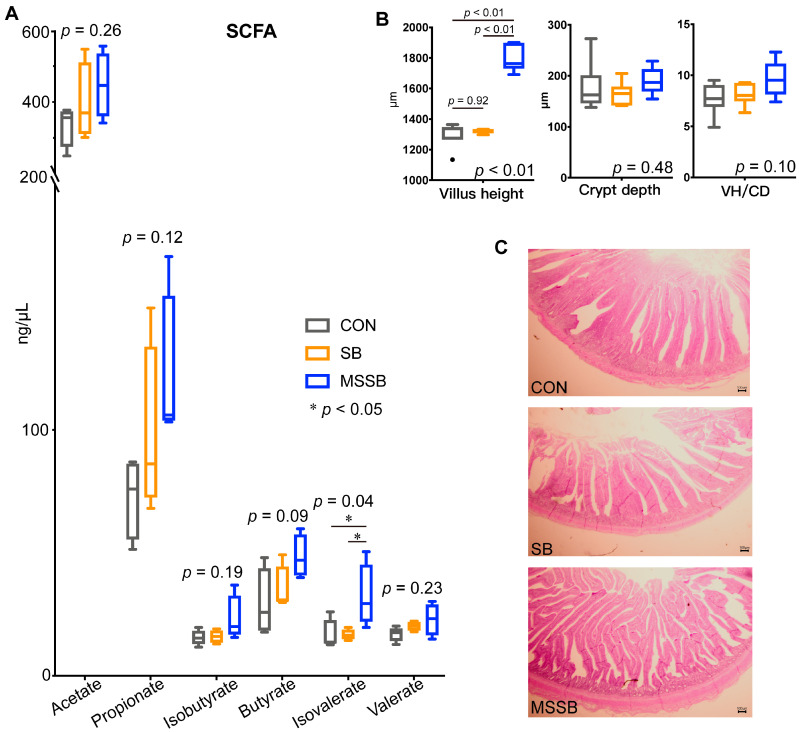
The SCFAs and gut morphology between the three groups. (**A**) The gut SCFAs between three groups. (**B**) The villus height, crypt depth, and the ratio of villus to crypt depth (VH/CD) between the three groups. (**C**) The morphology of jejunum between CON, SB, and MSSB. Abbreviations: CON, control; SB: sodium butyrate; MSSB: microcapsule sustained-release sodium butyrate.

**Figure 3 animals-13-03598-f003:**
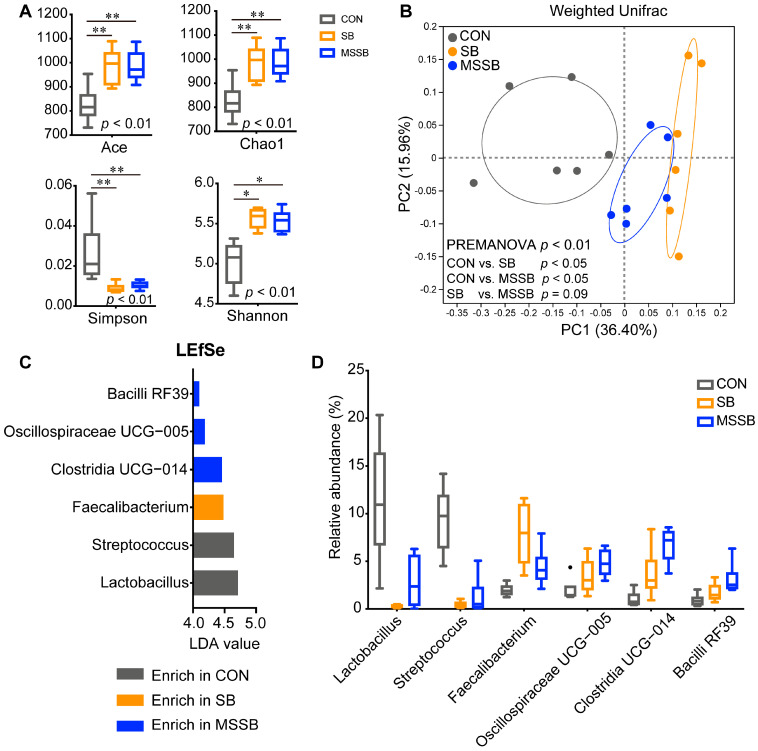
The gut microbiota between CON, SB, and MSSB groups. (**A**) The alpha diversity of gut microbiota. (**B**) The PcoA plot of ASV level between the three groups based on the weighted UniFrac distance matrix. (**C**) Linear discriminant analysis effect size (LefSe) of genus between three groups with an LDA value over 4.0 shown in this figure. (**D**) The relative abundance of differential genera in the three groups selected by LEfSe. Abbreviations: CON, control; SB: sodium butyrate; MSSB: microcapsule sustained-release sodium butyrate. * *p* < 0.05, ** *p* < 0.01.

**Figure 4 animals-13-03598-f004:**
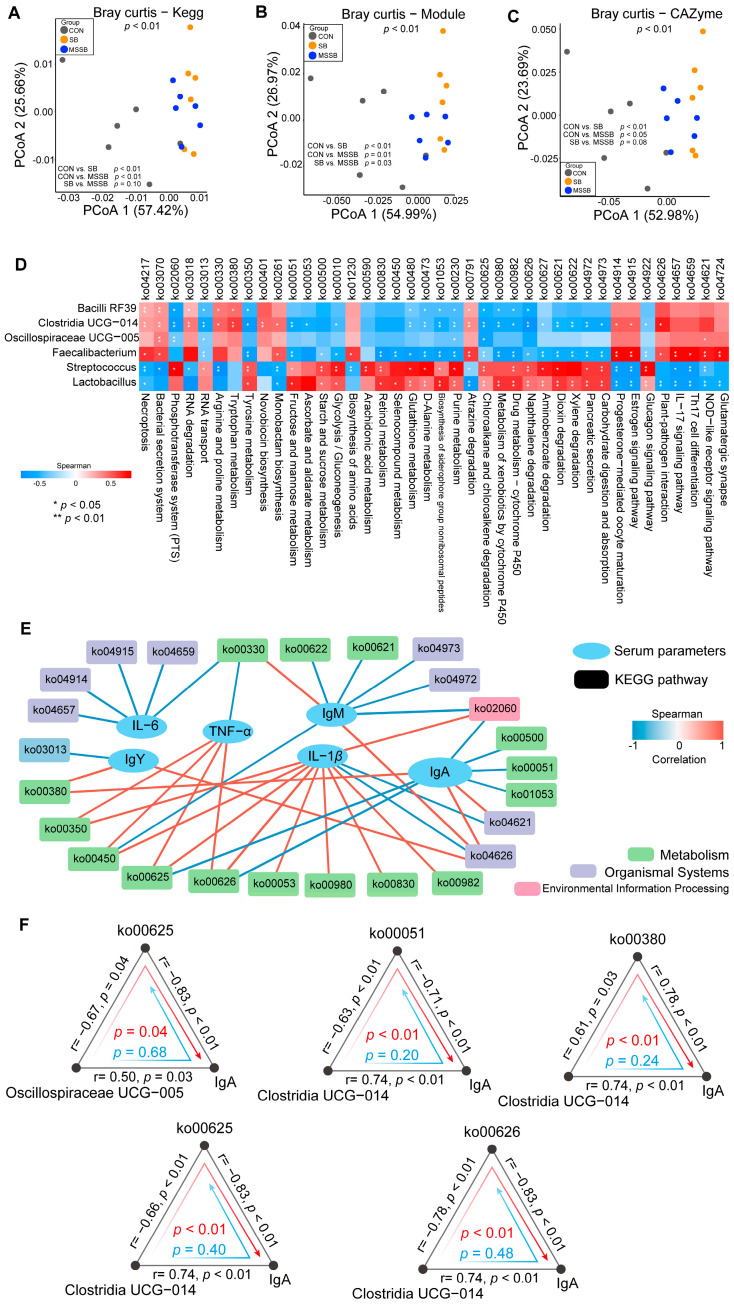
The functions of gut microbiota predicted by PICRUST2 and the causal effects between gut microbiota and immunity in yellow broilers administrated with dietary microcapsule sustained-release sodium butyrate. PCoA based on the Bray–Curtis distance matrix at (**A**) KEGG pathways, (**B**) modules, and (**C**) CAZymes of the gut microbiota between the three groups. (**D**) Correlation between differential genera (LDA score > 4) and differential pathways (*p* < 0.05). (**E**) The network between pathways and differential serum parameters, with Spearman correlation |r| over 0.7 and *p* value less than 0.05, is shown. (**F**) The mediation analysis identifies the linkages between the gut microbiota, pathways, and IgA. The gray lines represent the associations between the factors, with Spearman coefficients and *p* values shown. The red arrow represents the direct mediation and the blue arrow represents a reverse mediation, also with corresponding *p* values of mediation analysis shown in the figure.

**Table 1 animals-13-03598-t001:** The ingredients and nutrient levels of the basal diets applied in this study (as-fed basis).

Items (% Unless Noted)	Contents
Ingredients	
Corn	54.4
Soybean meal	23.6
Extruded full-fat soybean	5
Rice distiller’s grains	5
Soybean oil	2.2
Limestone	1.5
Fermented soybean meal	2.5
Corn protein meal	2.0
CaHPO_4_	2.0
NaCl	0.3
Premix ^(1)^	1.5
Total	100.00
Nutrient levels	
ME ^(2)^ (kcal/kg)	2983
CP	20.4
Lys	1.18
Met	0.55
Met+Cys	0.90
Try	0.22
Thr	0.88
Ca	0.86
TP ^(3)^	0.70
Non-phosphate	0.43

^(1)^ The premix provided the following per kg of diet: VA 10,000 IU, VB_1_ 1.5 mg, VB_2_ 3.5 mg, VB_6_ 3 mg, VB_12_ 10 μg, VD_3_ 2500 IU, VE 20 mg, VK 3.5 mg, biotin 0.15 mg, folic acid 1.0 mg, *D*-pantothenic acid 10 mg, nicotinic acid 30 mg, choline chloride 1000 mg, Cu (as copper sulfate) 8 mg, Fe (as ferrous sulfate) 80 mg, Mn (as manganese sulfate) 60 mg, Zn (as zinc sulfate) 60 mg, I (as potassium iodide) 0.18 mg, Se (as sodium selenite) 0.15 mg; ^(2)^ ME was a calculated value, while the others were measured values; ^(3)^ TP: total phosphate.

**Table 2 animals-13-03598-t002:** The effect of sodium butyrate and microcapsule sustained-release sodium butyrate on the performance of yellow broilers (mean ± SEM).

Items	CON	SB	MSSB	*p*-Value
ADG (g)	34.44 ± 1.46	37.42 ± 4.17	38.27 ±1.58	0.13
ADFI (g)	69.56 ± 3.52 ^b^	73.54 ± 0.83 ^ab^	74.50 ± 1.45 ^a^	0.02
F:G	2.37 ± 0.08 ^a^	2.23 ± 0.06 ^b^	2.25 ± 0.04 ^b^	<0.01

Significant differences by one-way ANOVA test and Fisher’s LSD are indicated by different letters, whereas an average with various superscripts in the column shows significant difference (*p* < 0.05). Abbreviations: ADG, average daily gain; ADFI: average daily feed intake (ADFI), and feed:gain ratio (F:G); CON, control; SB: sodium butyrate; MSSB: microcapsule sustained-release sodium butyrate.

## Data Availability

The DNA sequences of this study were deposited in the National Center for Biotechnology Information (NCBI) Sequence Read Archive (SRA) repository under accession number PRJNA1018725.

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
