# Peer review of "Effects of Dietary Microcapsule Sustained-Release Sodium Butyrate on the Growth Performance, Immunity, and Gut Microbiota of Yellow Broilers"

_animals, 2023, doi:10.3390/ani13233598_

Round 1
Reviewer 1 Report
Comments and Suggestions for Authors
* Comments and Suggestions for Authors
This manuscript is very interesting. I recommend acceptance after minor revisions.
1-line 14: dose not does
2-line 14: please use another terminology for SB such as rapid SB absorption
3-line 17-23: please highlight whether these findings are considerable (significant) or not.
4-line 31: please adjust the language.
5-line 66: please refer to the animal model and the target tissue or biofluids.
6-line 74-86: the train of information in this paragraph is not organized well. Please try to improve the rationale of this study.
7-line 89-91: please provide the ethical code
8-line 96-97: the dose was selected based on what? The authors are required to answer why they selected this dose.
9-line 112-119: the authors did samplings on only day 52. Why did they select this time point? Also, why they did not assess the parameters at different times during the study.
10-line 125-132: these biomarkers were not highlighted in the introduction section. The authors should highlight why they assess these biomarkers.
11- Also, please state the sensitivity and specificities of these kits for measurements.
12- line 143-144: why the authors targeted metabolomic assessment for specific ones.
13-line 150-165: Why the authors did not assess the parameters on different time points that could enhance the mechanism by which the different groups could affect the outcomes of the studied parameters.
14-line 166-188: what about the normality of your data?
15-table 2 why the data is presented in mean ±sem in all groups.
16-line 280: the discussion contains subtitles. Is it ok?
17-line 290-314: the authors should explain the findings. I found many sectences just confirm the findings of this study. Please try to explain your findings in the whole mechanism.
Comments on the Quality of English Language
Author Response
Reviewer1
Comments and Suggestions for Authors
This manuscript is very interesting. I recommend acceptance after minor revisions.
RE: We thank the reviewer for thoughtful review of our manuscript and for the recommendations to improve of our text. Our specific responses to each comment are in blue text in this response document.
1-line 14: dose not does
RE: Revised.
2-line 14: please use another terminology for SB such as rapid SB absorption
RE: Revised.
3-line 17-23: please highlight whether these findings are considerable (significant) or not.
RE: Thanks for your comment. We think these findings are considerable, especially they revealed the potential mechanism of the MSSB in the performance and immunity of yellows broilers. It has been highlighted in the manuscript in Line 24-25. Hope you can take it into consideration.
4-line 31: please adjust the language.
RE: Thanks for your comment. We have adjusted the language and highlighted in the manuscript (Line 33-34).
5-line 66: please refer to the animal model and the target tissue or biofluids.
RE: Thanks for your comment. The animal model and the target tissue has been referred (Line 68-69).
6-line 74-86: the train of information in this paragraph is not organized well. Please try to improve the rationale of this study.
RE: Thanks for your comment. The paragraph has been reorganized and improved (Line 77-89). Hope you can take it into consideration.
7-line 89-91: please provide the ethical code
RE: Thanks for your comment. The ethical code is ZAFU20230262, which has been added and highlighted in the manuscript.
8-line 96-97: the dose was selected based on what? The authors are required to answer why they selected this dose.
RE: Thanks for your comment. The dose was selected based on the previous study, which exert the best effect in the improvement of the growth performance in broilers. It has been added in the material and methods part (Experimental design and animal management). Hope you can take it into consideration.
9-line 112-119: the authors did samplings on only day 52. Why did they select this time point? Also, why they did not assess the parameters at different times during the study.
RE: Thanks for your comment. It takes 52 days from hatch to finish of yellow broilers. That’s why we choose day 52 as sampling date. The parameters such as the metabolites and morphology of gut assessed can only be assessed after the slaughter. So, it is unfortunate that the data on other time point were missed.
10-line 125-132: these biomarkers were not highlighted in the introduction section. The authors should highlight why they assess these biomarkers.
RE: Thanks for your comment. We have highlighted the biomarkers in the introduction part. Hope you can take it into consideration.
11- Also, please state the sensitivity and specificities of these kits for measurements.
RE: Thanks for your comment. The sensitivity and specificities of these kits were added in the materials and methods part.
12- line 143-144: why the authors targeted metabolomic assessment for specific ones.
RE: Thanks for your comment. We targeted the SCFA in the gut because the SCFA are the main metabolites produced in the gut and are considered to alleviate inflammation. We have added the importance of the SCFAs in the discussion part. That’s the reason why the SCFAs were assessed in our study. Also, assessment of other metabolites will be on schedule in our future work. Hope you can take it into consideration.
13-line 150-165: Why the authors did not assess the parameters on different time points that could enhance the mechanism by which the different groups could affect the outcomes of the studied parameters.
RE: Thanks for your comments. As mentioned before, the study was designed to evaluate the effects of MSSB on yellow broilers at finishing-point. Based on the results and your valuable suggestions, more time points can be set in future work for better evaluating the mechanism behind the beneficial effects of MSSB on yellow broilers.
14-line 166-188: what about the normality of your data?
RE: Thanks for your comments. The normality of our data has been detected with Shapiro-Wilk test in R. It has been modified in the statistical analysis part.
15-table 2 why the data is presented in mean ±sem in all groups.
RE: Thanks for your comments. The sem in all groups reflect the in our whole samples, also, the table2 has been modified in mean ±sem in each group in the manuscript. Hope you can take it into consideration.
16-line 280: the discussion contains subtitles. Is it ok?
RE: Thanks for your comments. The discussion contains subtitles could provide a clearer information and better understanding for readers, which makes our discussion more logical. That’s why the discussion contains subtitles. Hope you can take it into consideration.
17-line 290-314: the authors should explain the findings. I found many sectences just confirm the findings of this study. Please try to explain your findings in the whole mechanism.
RE: Thanks for your suggestions. Our study was designed to compare the difference between the SB and MSSB and several parameters including antioxidant capacity and cytokine showed same trend both in SB and MSSB. Another reason is the mechanism of both SB and MSSB in the immunomodulatory effects in broilers has not been fully described, further study is warrant to illustrate the whole mechanisms. So, we just simply confirm the findings of our study in this part. The further mechanism was discussed in the section “MSSB improve the IgA level potentially via the tryptophan metabolism in Clostridia UCG-014”.
Also, we are happy to further explain the mechanisms of our findings and has been highlighted in this part. Hope you can take it into consideration.

Reviewer 2 Report
Comments and Suggestions for Authors
The paper is relevant, that could bring novels insights about the butyrate.
The propose of this study has a good basement, with a methodology well established.
The study was well designed and conducted.
The results found were relevant and important for science and industry.
Presentation of some results could be improved.
Sometimes the results were confused, but I think that it’s possible to re-write.
Table 1. - to check: Try??? 0.22… TP??? 0.70…
Table 2. to check: Duncan’s multiple comparison -
It is not the best statistical test for this study.
You have three treatments and 2 least significant difference.
Inevitably, your interpretation is biased.
Prefer a test that compares treatments while maintaining a single LSD.
In Table 2: Include the general average before SEM.
Table 2 - Abbreviations: to check – Guideline – Animals
Abbreviations: it is common to use in the table footer
Include - significant differences by XXXX test are indicated by different letters, in line….
Th results were displayed in the form of the boxplot and standard error of means.
This is an excellent option.
On the other hand, the presence of influent values or outliers in the analyzed data is visible.
Therefore, I recommend using the homoscedasticity test to check whether the variations are heterogeneous.
If the variability of the animal response is due to genetic potential, the authors should dedicate space in the text to contextualize the animal model used in this research.
The amount of daily weight gain is strange to say the least.
Author Response
Reviewer2
Comments and Suggestions for Authors
The paper is relevant, that could bring novels insights about the butyrate.
The propose of this study has a good basement, with a methodology well established.
The study was well designed and conducted.
The results found were relevant and important for science and industry.
Presentation of some results could be improved.
Sometimes the results were confused, but I think that it’s possible to re-write.
RE: We thank the reviewer for thoughtful review of our manuscript and for the recommendations to improve of our text. Our specific responses to each comment are in blue text in this response document.
Table 1. - to check: Try??? 0.22… TP??? 0.70…
RE: Thanks for your comment. The Try is 0.22, and the TP is total phosphate, which are within a reasonable range. We have added the note below the table 1. Hope you can take it into your consideration.
Table 2. to check: Duncan’s multiple comparison -
It is not the best statistical test for this study.
You have three treatments and 2 least significant difference.
Inevitably, your interpretation is biased.
Prefer a test that compares treatments while maintaining a single LSD.
RE: Thanks for your comment. The PostHocTest LSD has been performed and has been modified in the statistical analysis part. Hope you can take it into your consideration.
In Table 2: Include the general average before SEM.
Table 2 - Abbreviations: to check – Guideline – Animals
Abbreviations: it is common to use in the table footer
Include - significant differences by XXXX test are indicated by different letters, in line….
RE: Thanks for your suggestion. The table has been modified. The abbreviations has been used in the table footer.
Th results were displayed in the form of the boxplot and standard error of means.
This is an excellent option.
On the other hand, the presence of influent values or outliers in the analyzed data is visible.
Therefore, I recommend using the homoscedasticity test to check whether the variations are heterogeneous.
RE: Thanks for your suggestion. The Breusch-Pagan test was applied to assess the homoscedasticity of the data, which has been modified in the statistical analysis part.
If the variability of the animal response is due to genetic potential, the authors should dedicate space in the text to contextualize the animal model used in this research.
RE: Thanks for your suggestion. The space has been dedicated for the yellow broiler in the introduction part, the animal model used in this research. Hope you can take it into consideration.
The amount of daily weight gain is strange to say the least.
RE: Thanks for your comment. The daily weight gain of yellow broiler with rapid growth rate, ranging from 30 - 40 g/bird, our results were in accordance with the previous report (https://doi.org/10.3382/ps/peu003). Hope you can take it into consideration.

Reviewer 3 Report
Comments and Suggestions for Authors
Line 11-12: you meant something like this? “However, the rapid release of SB in the gastrointestinal tract leads to lower efficiency in poultry production”
Line 14: instead of does dose? "There are several errors in the text; it would be advisable to have it reviewed."
The choice of the 1000 mg/kg additive concentration was made as it is roughly three times higher than the typical dosage. This decision was based on specific research or considerations, but it raises questions about its sustainability. There are potential environmental and animal health implications associated with this higher concentration. Further evaluation is needed to assess the long-term viability and impact on both the environment and animal health.
It would have been better to examine the parameters in various stages of growth across different sectors, not just at the end of the experiment. It would have been beneficial to monitor the changes in redox homeostasis throughout the experiment, as it is a highly dynamic and mutually potentiating system.
The choice to focus on specific small-molecule antioxidant compounds for examination, without considering other alternatives, raises the question of why these particular compounds were selected.
Line 153: Which 16S regions are you examining with these PCR primers?
Line 158- 160: What parameters did you use for DADA2 pipeline that resulted the 6,684 ASV?
Line 169 -170: In which program did you run LEfSe analysis? You should add a citation here about LEfSe analysis.
Comments on the Quality of English LanguageLine 11-12: you meant something like this? “However, the rapid release of SB in the gastrointestinal tract leads to lower efficiency in poultry production”
Line 14: instead of does dose? "There are several errors in the text; it would be advisable to have it reviewed."
The choice of the 1000 mg/kg additive concentration was made as it is roughly three times higher than the typical dosage. This decision was based on specific research or considerations, but it raises questions about its sustainability. There are potential environmental and animal health implications associated with this higher concentration. Further evaluation is needed to assess the long-term viability and impact on both the environment and animal health.
It would have been better to examine the parameters in various stages of growth across different sectors, not just at the end of the experiment. It would have been beneficial to monitor the changes in redox homeostasis throughout the experiment, as it is a highly dynamic and mutually potentiating system.
The choice to focus on specific small-molecule antioxidant compounds for examination, without considering other alternatives, raises the question of why these particular compounds were selected.
Line 153: Which 16S regions are you examining with these PCR primers?
Line 158- 160: What parameters did you use for DADA2 pipeline that resulted the 6,684 ASV?
Line 169 -170: In which program did you run LEfSe analysis? You should add a citation here about LEfSe analysis.
Author Response
Reviewer3
Line 11-12: you meant something like this? “However, the rapid release of SB in the gastrointestinal tract leads to lower efficiency in poultry production”
RE: Thanks for your comment. The sentence has been modified and highlighted in the manuscript according to your advice.
Line 14: instead of does dose? "There are several errors in the text; it would be advisable to have it reviewed."
RE: Thanks for your comment. The “dose” has been revised throughout the manuscript.
The choice of the 1000 mg/kg additive concentration was made as it is roughly three times higher than the typical dosage. This decision was based on specific research or considerations, but it raises questions about its sustainability. There are potential environmental and animal health implications associated with this higher concentration. Further evaluation is needed to assess the long-term viability and impact on both the environment and animal health.
RE: Thanks for your comment. Compared to the 1000 mg/kg SB, the MSSB only contains 40% of SB, reducing the usage of SB and better effects than the high amount of SB. Also, just like the reviewer said, the long-term effect on both the environment and animal health warrant further assessed. Hope you can take it into consideration.
It would have been better to examine the parameters in various stages of growth across different sectors, not just at the end of the experiment. It would have been beneficial to monitor the changes in redox homeostasis throughout the experiment, as it is a highly dynamic and mutually potentiating system.
RE: Thanks for your valuable suggestion. Our study was intended to assess the performance at the end point. Also, just like you said, more time points can be set in future work for better evaluating the mechanism behind the beneficial effects of MSSB on yellow broilers.
The choice to focus on specific small-molecule antioxidant compounds for examination, without considering other alternatives, raises the question of why these particular compounds were selected.
RE: Thanks for your valuable suggestion. The antioxidant compounds were representative for assessment of the antioxidant levels. That’s the reason why we choose the antioxidant compounds for examination. The introduction of these compounds were add in the background part. Hope you can take it into your consideration.
Line 153: Which 16S regions are you examining with these PCR primers?
RE: Thanks for your valuable suggestion. The V3-V4 region was examined with primers, which has been added in the 16S rRNA gene sequencing part.
Line 158- 160: What parameters did you use for DADA2 pipeline that resulted the 6,684 ASV?
RE: Thanks for your valuable suggestion. The parameters of DADA2 pipeline was below:
--p-trim-left-f 0 \
--p-trim-left-r 0 \
--p-trunc-len-f 230 \
--p-trunc-len-r 230 \
The parameters were set according to the quality of the initial bases. We didn’t trim the bases from the beginning of the sequences.
Line 169 -170: In which program did you run LEfSe analysis? You should add a citation here about LEfSe analysis.
RE: Thanks for your comment. the LEfSe was performed in R with “lefser” package, and the citation was add in the statistical analysis part. Hope you can take it into consideration.

Reviewer 4 Report
Comments and Suggestions for Authors
The manuscript is interesting and can be published because it contributes to science. However, adjustments will be necessary:
a) the term supplementation was used incorrectly in the title and text. What does supplement mean? it is offering something extra, something that is already in the diet; that's not the case here. adjust all text.
b) the conclusion of the "abstract" section needs to be reformulated; remembering that it must respond to the main objective. In the present formed, it remained somewhat speculative.
c) in table 1, was the chemical composition of the diet presented in a calculated or analyzed way? The ideal would be to have both, even to make sure that the diets produced had the same composition.
d) the statistical analysis of the microbiota was very superficial, needs more details, including the database used, the form analyzed, etc.
e) I have to praise the neatness of the figures, of excellent quality and with an easy-to-see data presentation. congratulations
f) find a way to edit so that page 9 does not run out of information
g) both in the introduction and in the discussion, I missed the authors making it clear why (justification) for microencapsulating butyrate? make this more clear.
h) The conclusion is appropriate; but I believe it could be more elaborate; but I could have better valued the excellent results obtained.
i) articles with butyric acid glycerides have been published in the last 2 years in broiler chickens, it is strange that these articles are not part of the text (they could support the introduction and discussion). I understand that these articles need to be cited; just as authors need to make it clear what the differences between the studies are; the knowledge gap that still exists. I clearly understand that the study presented here complements the others. The study by Ficagna et al. 2023 is just one example.
Author Response
Reviewer4
Comments and Suggestions for Authors
The manuscript is interesting and can be published because it contributes to science. However, adjustments will be necessary:
- a) the term supplementation was used incorrectly in the title and text. What does supplement mean? it is offering something extra, something that is already in the diet; that's not the case here. adjust all text.
RE: Thanks for your comment. The term “supplementation” has been adjusted.
- b) the conclusion of the "abstract" section needs to be reformulated; remembering that it must respond to the main objective. In the present formed, it remained somewhat speculative.
RE: Thanks for your comment. The conclusion of the “abstract” has been reformulated and avoid speculative.
- c) in table 1, was the chemical composition of the diet presented in a calculated or analyzed way? The ideal would be to have both, even to make sure that the diets produced had the same composition.
RE: Thanks for your comment. The chemical composition of the diet was presented in a calculated way. The chemical composition was analyzed from the dietary sample, which truly reflect the composition of the diet.
- d) the statistical analysis of the microbiota was very superficial, needs more details, including the database used, the form analyzed, etc.
RE: Thanks for your comment. The analysis of microbiota in detail was in the 16S rRNA gene sequencing part, and the databases used in the analysis were added in this part. Hope you can take it into consideration.
- e) I have to praise the neatness of the figures, of excellent quality and with an easy-to-see data presentation. Congratulations
RE: Thanks for your comment.
- f) find a way to edit so that page 9 does not run out of information
RE: Thanks for your comment. The page 9 has been modified. Hope you can take it into consideration.
- g) both in the introduction and in the discussion, I missed the authors making it clear why (justification) for microencapsulating butyrate? make this more clear.
RE: Thanks for your comment. The microencapsulating butyrate was able to effectively deliver the butyrate throughout the entire gastrointestinal tract. On the other hand, the microencapsulating butyrate contains only 40% of SB but exert better beneficial effects than SB. We have made the reason why microencapsulating butyrate more clear and highlighted in the manuscript. Hope you can take it into consideration.
- h) The conclusion is appropriate; but I believe it could be more elaborate; but I could have better valued the excellent results obtained.
RE: Thanks for your comment. The conclusion has been improved to be more elaborate. Hope you can take it into consideration.
- i) articles with butyric acid glycerides have been published in the last 2 years in broiler chickens, it is strange that these articles are not part of the text (they could support the introduction and discussion). I understand that these articles need to be cited; just as authors need to make it clear what the differences between the studies are; the knowledge gap that still exists. I clearly understand that the study presented here complements the others. The study by Ficagna et al. 2023 is just one example.
RE: Thanks for your comment. the study with butyric acid glycerides has been cited both in background and discussion part. Hope you can take it into consideration.

Round 2
Reviewer 1 Report
Comments and Suggestions for Authors
Thank you for addressing the comments in the first Round of revision. I noticed the quality of some figures is not good. Please try to improve the quality of uploaded figures (especially figure 4).
Comments on the Quality of English LanguageMinor revision for the language.
Reviewer 4 Report
Comments and Suggestions for Authors
Adjusted were done.